# Immunotherapy as a Turning Point in the Treatment of Acute Myeloid Leukemia

**DOI:** 10.3390/cancers13246246

**Published:** 2021-12-13

**Authors:** Anna Aureli, Beatrice Marziani, Tommaso Sconocchia, Maria Ilaria Del Principe, Elisa Buzzatti, Gianmario Pasqualone, Adriano Venditti, Giuseppe Sconocchia

**Affiliations:** 1CNR Institute of Translational Pharmacology, 00133 Rome, Italy; 2Emergency and Urgent Department, University Hospital Sant’Anna of Ferrara, 44124 Ferrara, Italy; mrzbrc@unife.it; 3Division of Hematology, Medical University of Graz, 8036 Graz, Austria; tommaso.sconocchia@medunigraz.at; 4Hematology, Department of Biomedicine and Prevention, University Tor Vergata, 00133 Rome, Italy; del.principe@med.uniroma2.it (M.I.D.P.); elisa.buzzatti@ptvonline.it (E.B.); gianmarco.pasqualone@ptvonline.it (G.P.); adriano.venditti@uniroma2.it (A.V.)

**Keywords:** AML, immunotherapy, antibody, antibody–drug conjugate, targeted therapies

## Abstract

**Simple Summary:**

Despite recent progress achieved in the management of acute myeloid leukemia (AML), it remains a life-threatening disease with a poor prognosis, particularly in the elderly, having an average 5-year survival of approximately 28%. However, recent evidence suggests that immunotherapy can provide the background for developing personalized targeted therapy to improve the clinical course of AML patients. Our review aimed to assess the immunotherapy effectiveness in AML by discussing the impact of monoclonal antibodies, immune checkpoint inhibitors, chimeric antigen receptor T cells, and vaccines in AML preclinical and clinical studies.

**Abstract:**

Acute myeloid leukemia (AML) is a malignant disease of hematopoietic precursors at the earliest stage of maturation, resulting in a clonalproliferation of myoblasts replacing normal hematopoiesis. AML represents one of the most common types of leukemia, mostly affecting elderly patients. To date, standard chemotherapy protocols are only effective in patients at low risk of relapse and therapy-related mortality. The average 5-year overall survival (OS) is approximately 28%. Allogeneic hematopoietic stem cell transplantation (HSCT) improves prognosis but is limited by donor availability, a relatively young age of patients, and absence of significant comorbidities. Moreover, it is associated with significant morbidity and mortality. However, increasing understanding of AML immunobiology is leading to the development of innovative therapeutic strategies. Immunotherapy is considered an attractive strategy for controlling and eliminating the disease. It can be a real breakthrough in the treatment of leukemia, especially in patients who are not eligible forintensive chemotherapy. In this review, we focused on the progress of immunotherapy in the field of AML by discussing monoclonal antibodies (mAbs), immune checkpoint inhibitors, chimeric antigen receptor T cells (CAR-T cells), and vaccine therapeutic choices.

## 1. Introduction

Acute myeloid leukemia (AML) is the leading cause of mortality among all leukemias [1].

It starts in the bone marrow (BM), moves into the blood, and sometimes spreads to other organs including the lymph nodes, liver, spleen, skin, testicles, and central nervous system [2,3]. AML is generally characterized by cytogenetic and genetic aberrations that alter the normal hematopoietic growth and the differentiation of progenitor cells, resulting in bone marrow failure and systemic blast cell dissemination in the peripheral blood [4].

In the past, a one-size-fits-all regimen of either high- or low-intensity chemotherapy combined with HSCT was the only available treatment for AML. Response to this treatment is not always fully satisfactory. Hematological remission is only obtained in about a third of patients. This percentage is higher in patients under the age of 60–65, with success rates approaching 50–60%, while, for older patients, long-term survival beyond 5 years does not exceed 10–20% of cases [5,6]. Note that the survival improvement in younger patients has been credited to supportive care and improved HSCT techniques rather than an improvement in pharmacotherapies. The achievement of significant progress in the management of AML is mainly due to a better comprehension of the genetic and molecular mechanisms underlying the clinical course of the disease. In contrast, the real impact of immunotherapeutic approaches is not clearly defined.

The characterization of human tumor antigens [7], the introduction of therapeutic monoclonal antibodies (mAbs) in clinical oncology [8], and the elucidation of the role of immunological checkpoint inhibitors in preventing effective antitumor immune responses provide investigators with an array of therapeutic tools to be utilized as a platform for designing rational immunotherapy strategies for AML [9].

In acute lymphoblastic leukemia (ALL), immunotherapy is extensively utilized, and defined antibody-based approaches are already included in standard protocols [10]. In contrast, in AML, more specific cell surface targets have not yet been identified [11,12].

Although HSCT remains the most effective treatment, AML relapse can still occur. In addition, most AML patients are elderly and therefore are not suitable for this type of treatment. Leukemic stem cells (LSCs), resistant to chemotherapy and radiotherapy, are supposed to be responsible for the minimal residual disease (MRD) that predicts relapse but may not be a contraindication for HSCT [13], and whose monitoring remains unsatisfactory [14]. Thus, there is a need to develop alternative strategies resulting in long-term remission with minimal toxicity also for patients who are not eligible for current treatments [15].

A variety of treatment protocols for AML, based on immune-mediated therapeutic mechanisms, have been developed in recent years. Here, we will provide a critical overview of the most important immunotherapeutic treatments for AML.

## 2. Antibody-Based Therapy

The current concept of antibody-based therapy began with Porter’s finding which disclosed the basic structure of immunoglobulin (Ig). He showed that papain digestion of a rabbit antibody produced two antigen-binding fragments (Fab) and a third fragment that is easily crystallizable (Fc). The former interfered with the antigen binding of the undigested antibody, while the latter did not [16]. To date, antibody-based therapies have been aimed at detecting and selectively destroying target malignant cells by different mechanisms:(1)antibody–drug conjugated to various toxins that, following cell surface binding to the antigen, are first internalized, then released in the lysosomes, and finally delivered to the nucleus where they induce cell death through the DNA double-strand breaks and cell cycle arrest [17]; (2)bispecific antibody (BsAb) which results from the fusion of an antibody specific for a triggering molecule on the effector cell, and an antibody specific for a cell surface tumor-associated antigen (TAA) on malignant cells [18]; (3)naked antibody mediating antibody-dependent cellular cytotoxicity (ADCC) and complement-mediated cytotoxicity (CDC), and/or interfering with inhibitory checkpoints [19].

### 2.1. Antibody–Drug Conjugates (ADCs)

#### 2.1.1. CD33

CD33 is a sialic acid binding receptor highly expressed on myeloid cells [20]. It is rapidly internalized, when engaged with antibodies, and this makes it an interesting target for conjugated antibody–drug therapy. Various ADCs have been designed including gemtuzumabozogamicin (GO; Mylotarg™). This molecule is composed of a humanized anti-CD33 monoclonal antibody (mAb) conjugated to a cytotoxic agent such as calicheamicin [21]. GO was the first and most promising anti-CD33 mAb to obtain accelerated approval in 2000 by the Food and Drug Administration (FDA) for its use in first-relapse AML in patients not eligible for conventional chemotherapy [22]. Nevertheless, the approval was dependent on its validation in prospective randomized trials. Thus, GO was evaluated in numerous clinical studies which included different stages of the disease, in monotherapy or in combination with chemotherapy [23,24,25]. Importantly, as highlighted in the phase III randomized multicenter clinical trial NCT00085709, which compared GO 6 mg/m^2^ on day 4 plus a daunorubicin and cytarabine (DA) induction chemotherapy regimen to that of standard DA, patients receiving GO had a higher mortality rate, due to venous occlusive disease (VOD) [26,27]. Moreover, the indication that the addition of GO to induction or maintenance therapy failed to improve the complete response (CR) rate or overall survival (OS) in patients with AML led to a voluntary withdrawal [26].

However, based on preliminary studies showing that implementation of lower doses of GO (3 mg/m^2^) can be safely associated with induction chemotherapy regimens [28], two major clinical trials were carried out. The UK MRC AML15 [29] and UK NCRI AML16 [30] trials included patients younger than 60 years and older than 60 years, respectively. The former study was a vast randomized trial during induction/consolidation chemotherapy. The study investigated GO in non-PML AML through all groups of risk. Patients received one of the following chemotherapeutic regimens: DA; fludarabine, cytarabine, granulocyte colony-stimulating factor (G-CSF), and idarubicin (FLAG-Ida); cytarabine, daunorubicin, and etoposide (ADE), with or without 3 mg/m^2^ of GO on day 1 induction. GO was well tolerated; karyotypic analysis indicated a significant interaction with GO and enhanced survival of patients with favorable cytogenetics. The latter was a randomized study involving untreated AML or MDS patients who were given DA or daunorubicin and clofarabine (DC) with or without GO (3 mg/m^2^) on day 1 of induction therapy. No toxicity was observed in the presence of GO, while there was a better 3-year cumulative incidence of relapse and survival.

Further investigations have evaluated the potential benefit of GO addition to induction or maintenance therapy, but no improved survival data in AML patients have been demonstrated.

The GOELAMS AML 20061R study was a phase III clinical trial testing the impact of GO’s association with standard chemotherapy, focusing on patients with an intermediate karyotype. The study included 238 AML patients between 18 and 60 years of age who were randomized to standard DA plus or minus GO 6 mg/m^2^. There was significantly higher hepatotoxicity in the GO regimen and four cases of VOD. However, the event-free survival (EFS) was significantly higher in the GO regimen. 

The ALFA-0701 [31] study was an additional study in which AML patients aged 50–70 years received the standard, frontline 3 + 7 (AD) chemotherapy. This study was a multicenter, phase III clinical trial involving 280 patients, 140 of whom received intravenous GO (3 mg/m^2^, maximum dose of 5 mg) on days 1, 4, and 7. GO had no impact on the number of patients achieving CR. However, in the following 2 years, the EFS, OS, and relapse-free survival (RFS) of patients treated with GO were significantly better than without GO. In contrast, hematologic toxicity, and particularly thrombocytopenia, was more common with GO. The study suggested that fractionation of a low dose of GO is safe and improves the clinical course of AML. 

Since the results obtained from randomized phase III trials were not conclusive, a meta-analysis of individual patients included in the five above-described studies was carried out. GO did not increase the CR rate, while it significantly improved 5–6-year OS and significantly decreased the risk of relapse in patients with favorable and intermediate-risk cytogenetics [32].

In 2017, GO was re-approved for newly diagnosed and relapsed/refractory (R/R) AML owing to new data on the clinical efficacy and safety of GO administered according to a fractionated dosing schedule [33].

Up to that date, several clinical trials were carried out to study how to reduce GO’s toxicity. Many of these chose to enrich GO treatment with azacytidine. Azacytidine induced maturation of AML blasts, increased CD33 expression, and enhanced GO uptake by these cells [34]. This treatment can be used for induction and post-remission therapy in older patients with AML, allowing the achievement of a CR rate similar to that achieved with chemotherapy-based regimens [35].

An alternative approach is represented by the use of SGN-CD33A, a humanized anti-CD33 mAb conjugated to a new pyrrolobenzodiazepine dimer, whose safety and efficacy, alone or in various combinations, were evaluated in several clinical trials. Encouraging results by Sutherland et al. demonstrated that SGN-CD33A is highly active in a broad panel of preclinical AML models [36].

Interestingly, SGN-CD33 used as monotherapy shows a favorable antileukemic activity. In a phase I trial (NCT01902329), 14 out of 27 (54%) high-risk older AML patients achieved complete remission (CR) and incomplete blood count recovery (CRi), thereby more than doubling the response rate expected upon standard non-intensive therapies such as HMA or low-dose cytarabine. Furthermore, treatment with SGN-CD33A + HMA in older AML patients (NCT01902329) allowed reaching high rates of remission and protracted myelosuppression, with an encouraging CR/CRi rate of 73% [36]. It remains to be understood how potential hepatotoxicity and VOD can be controlled, which represents a major concern particularly in the combination of SGN-CD33A with allogenic HSCT before or after therapy(http://businesswire.com/Clinical-Hold-Phase1, accessed on 15 October 2021).

#### 2.1.2. CD123

Another reliable therapeutic target in the treatment of hematological malignancies is the interleukin 3 (IL-3) receptor α-chain (IL3RA) or CD123. It is strongly expressed on myeloid blasts and LSCs, but also in a small subset of CD56+ monocytes [37,38]. The CD123-directed immunoconjugate SGN-CD123A, developed by Seattle Genetics, is a humanized anti-CD123 antibody, conjugated to a powerful DNA binding pyrrolobenzodiazepine (PBD) dimer drug via a protease-cleavable dipeptide linker. It shows significant antitumor activity against a broad panel of primary AML samples and in preclinical models of MDR-positive AML that are characteristically resistant to chemotherapy [39]. Despite these interesting results, the phase I trial NCT02848248 was recently terminated because of serious concerns about its toxicity. Along with SGN-CD123A, the SL-101 molecule is composed of the anti-CD123 single chain (scFv) linked to a truncated Pseudomonas exotoxin deprived of its natural domain of targeting. This molecule has shown cytotoxic activity on laboratory and primary AML cells, and currently, a phase I study is being undertaken. Recently, however, an interesting fusion protein composed of IL-3 and a truncated version of the diphtheria toxin has been used. From this combination, SL-401 or tagraxofusp was generated, which reached phase II clinical evaluation, demonstrating a strong activity especially in patients with blastic plasmacytoid dendritic cell neoplasm (BPDCN), an aggressive hematologic malignancy that rapidly evolves to a leukemia phase and whose blasts overexpress IL-3R [40]. Additional results obtained from SL-401’s effectiveness investigation in AML patients were encouraging and promising since clear response rates were achieved in different myeloid malignancies, including the eradication of MRD. Its use, in combination with azacytidine, is also currently under investigation (NCT03113643). Important results were obtained in the NCT02113982 study involving untreated or relapsed BPDCN and R/R AML patients, who received intravenous tagraxofusp (7 μg or 12 μg/Kg) on days 1–5 every 3 weeks until progression or intolerable toxicity. The study showed that 57% of patients who did not receive a previous treatment achieved complete or minimal skin responses. In addition, 21 out of 65 patients could have stem cell transplantation, leading to prolongation of survival. The investigators concluded that the use of tagraxofusp led to significant clinical responses. However, important adverse effects were frequently identified including vascular leak syndrome, hepatic toxicity, and thrombocytopenia [41].

Despite the limited amount of information about this treatment, under exceptional circumstances, tagraxofusp was approved as an orphan medicine by the European Medicines Agency (EMA) in November 2015 (https://www.ema.europa.eu/en/medicines/human/EPAR/elzonris#authorisation-details-section, accessed on 8 November 2021), followed by the FDA authorization in December 2018. 

Figure 1 shows an overview of immunotargeting of validated targets in AML: approved versus investigational molecules.

### 2.2. Bispecific Antibodies

The first paper showing the description of the production of bispecific antibodies (BsAbs) was published in 1961 [42]. Then, the interest in BsAbs reached its peak in the 1980s and 1990s. Importantly, in 1985, two research teams in the Immunotargeting Section at the NIH (Bethesda) and in the Department of Immunology at Scripps (La Jolla) developed the first BsAbs composed of anti-CD3 heteroaggregates to target cells for which pertinent antibodies were available [43,44]. Then, this technology was constantly optimized [18]. The first BsAb used in hematologic malignancies was CD3xCD19, about 26 years ago, without a tangible clinical response in the clinical course of non-Hodgkin lymphoma (NHL) [45], while CD30xCD16 showed some clinical responses in Hodgkin lymphoma (HL) [46]. The current concept of antibody therapy was established when Rodney Porters disclosed another emerging typeof antibody-based therapy represented by BsAbs that comprise bispecific T cell engagers (BiTEs) and dual-affinity retargeting antibodies (DARTs). BiTE antibodies combine into one molecule with the specificities of two mAbs, one that binds a tumor-associated surface antigen and the other for surface proteins expressed on T cells or NK cells, triggering their effector potential; they exploit the cytotoxic activity of polyclonal T cells and cause highly efficient lysis of targeted tumor cells [47]. In 2004, the CD19xCD3 BiTE blinatumomab was implemented in a phase I trial, with some success.

The interesting data obtained with bispecific antibodies in ALL have prompted the development of the first bispecific antibody CD33/CD3BiTE, named AMG330 [48], for AML treatment. This BsAb recognizes the CD33 V-type domain and the T cell antigen CD3 [49]. Laszlo’s study showed that AMG330 mediates a powerful killing of human AML cells in vitro. Considering that AMG 330 activity is not affected by ABC transporter activity, and that CD33 expression is not decreased upon prolonged drug exposure, AMG330 may overcome important limitations of previous CD33-targeted therapeutics, including GO [20,48,50]. Following these interesting preclinical data, the phase I clinical trial NCT02520427 started in 2016, and preliminary results were published in 2018 at the ASH meeting. A total of 40 out of 60 patients (67%) enrolled in this phase I dose-escalation study had short periods of cytokine release syndrome (CRS) which responded well to treatment. Anyhow, there was a low frequency of remission induction, and high rates of disease progression [25,51,52].

The second BsAb format is represented by DART, in which heavy- and light-chain variable domains are on two separate polypeptides stabilized by a C-terminal disulfide bridge [53].

Flotetuzumab (FLZ), also known as MGD006 or S80880, is currently being investigated as a treatment in AML. This so-called DART simultaneously binds CD3 and CD123, leading T cells to recognize and kill in vitro and in vivo AML cell lines and primary AML blasts expressing CD123 [54]. An ongoing study(NCT02152956), which is being carried out in patients with AML and MDS that do not benefit from chemotherapy, is investigating the clinical effects and maximum tolerated dose of FLZ. The preliminary results show that it is well tolerated and exhibits considerable antileukemia activity [55,56]. Moreover, it has been shown that FLZ induces a clinical response in patients with refractory AML [55].

Together with AMG330 and Flotetuzumab, JNJ-63709178 (Janssen Pharmaceuticals) and MCLA-117 have also entered clinical trials for AML (Table 1). The first one is a CD3xCD123 bispecific IgG1 antibody generated using a process known as GenmabDuoBody^®^ technology, which, in contrast to BiTEs and DARTs, retains its Fc region and its associated effector functions and in vivo stability.

In addition to its safety and tolerability, a phase I study (NCT02715011) is evaluating the antitumor activity of JNJ-63709178 in R/R AML patients. Instead, the second one is a new T cell-redirecting antibody targeting CD3 on T cells and C-type lectin-like molecule-1 (CLL1) on leukemic cells [57]. This bispecific CLL1/CD3 antibody construct (MCLA-117) induces targeted antigen-specific cytotoxicity against primary AML cells [58]. Results published by van Loo et al. indicated that it could be a promising new T cell-mediated immunotherapy for all subtypes of AML.MCLA-117 efficiently redirects T cells to kill tumor cells while sparing normal HSCs. It is currently being evaluated in a phase I clinical trial (MCLA-117-CL01, NCT03038230) in R/R or elderly, previously untreated AML patients. However, the preliminary data demonstrate a limited clinical activity of MCLA-117: 26 out of the 50 patients treated were evaluable with a follow-up bone marrow assessment, and 4 patients showed greater than 50% blast reduction. No dose-limiting toxicity was observed.

### 2.3. Monoclonal Antibodies Directed against Human Leukemia Stem Cells (LSCs)

LSCs play an important role in promoting high relapse rates and treatment resistance in AML patients. LSCs are capable of self-renewal and, at the same time, continue to generate proliferating progenitors and leukemic blasts [59]. Therefore, targeted elimination of LSCs may represent a promising way to obtain prolonged remission in the absence of important side effects [60].

CLL-1 is one of the most studied targets that is involved in the regulation of key regulatory immune functions. Additionally, it is highly expressed in myeloid cells, AML blasts [57], and LSCs. CLL-1 is differentially distributed within the CD34+ cell compartment, being preferentially expressed on CD34+/CD38− AML blasts, while it is absent on normal CD34+/CD38− cells [61,62]. Thus, anti-CLL-1 could also be used for the detection of MRD, being not only a good therapeutic option but also a useful prognostic marker. Zhao et al. generated specific mAbs against CLL-1 and demonstrated their direct cytotoxic and anticancer activity in vitro and in vivo [63]. Furthermore, among LSC-associated surface antigens exploited for LSC-selective therapy, CD44, CD47, and CD123 are the best known [64].

CD44 is a transmembrane glycoprotein mainly expressed on hematopoietic cells. The gene is composed of 20 exons which give rise to a variety of isoforms: the standard isoform (CD44s), of about 85kDa [65], and the variant isoforms(CD44v) have a higher molecular weight, up to 250kDa [66]. CD44s comprises exons1–5 and 16–20. It is associated with several functions including cell–cell adhesion and migration. Additionally, it is involved in NK and NKTcell cytotoxicity [67], and that of polymorphonuclear cells, as well as in pro inflammatory cytokine production and cytotoxicity [68,69]. Interestingly, CD44v is implicated in myeloid leukemia pathogenesis. In vivo administration of an activating mAb specific for CD44 (H90) to NOD/SCID (non-obese diabetic/severe combined immune-deficient) mice, transplanted with human AML cells, considerably inhibited leukemic repopulation, indicating that AML LSCs are directly targeted [70]. A limitation of this treatment might be represented by its potential toxicity towards normal cells due to widespread target expression. Among CD44 variants, CD44v6 is associated with a poor survival rate in AML patients [71]. Thus, targeting CD44 variant isoforms can inhibit leukemia growth, as shown by Erb et al., who found that targeting CD44v10 prolonged the survival time in a mouse model of EL4 lymphoma [72].

CD47 is a cell surface protein belonging to the Ig superfamily, implicated in multiple cellular processes including protein–protein interactions [73]. It is upregulated in AML cells and plays an important role in promoting immune escape since it inhibits phagocytosis once bound to signal regulatory protein α (SIRPα) on macrophages [74]. In addition, SIRPα is expressed on hematopoietic cells and is involved in NK cell function [75,76]. Anti-CD47 antibodies blocking the CD47–SIRPα interaction permit macrophage-mediated phagocytosis of human AML LSCs [77]. Unfortunately, although to lesser extents, CD47 is also expressed on normal HSCs and progenitor cells [78,79]. Therefore, side effects such as hemolysis and anemia have been reported upon treatment [80,81]. Recent studies have benefittedfromHu5F9-G4, a monoclonal anti-CD47 antibody, used alone and in combination with azacytidine for R/RAML patients (NCT02678338, NCT03248479). Anemia was a dose-limiting adverse event [81]. Moreover, within this context, TTI-621 (SIRPαFc), a fusion protein formed by the N-terminal portion of SIRPα with the IgG1 Fc region, is currently being investigated. In phase I clinical trials, it has been shown that it does not provoke anemia in comparison to HU5F9-G4 due to minimal erythrocyte binding [82], but dose-dependent thrombocytopenia may occur, possibly caused by a phagocytic clearance of platelets [81]. Further clinical trials are ongoing to evaluate the efficacy of other therapeutic CD47/SIRPα antibodies for AML treatment [83]. Notably, the NCT02367196 open-label, phase I study was conducted in patients with R/R AML and high-risk MDS to demonstrate the efficacy of CC-90002 as monotherapy or in combination with rituximab. The CC-90002-AML-001 study was interrupted in dose escalation due to a lack of preliminary monotherapy activity and evidence of antidrug antibodies(ADAs) in most patients. CC-90002 used in combination with rituximab is being investigated in CD20+ NHL to enhance the efficacy of CD47 blockade while reducing ADAs [84]. In NCT03013218, ALX-148 (Evorpacept) is being explored as a single agent or in combination with pembrolizumab, trastuzumab, or rituximab in 30 patients with advanced solid tumors and lymphoma. The first clinical data show that it is well tolerated at the doses evaluated (0.3 mg/kg [mpk] IV every week [QW]—30 mpk every other week [QoW]). Therefore, the NCT04755244 study has also begun in 97 patients with AML to evaluate the safety and tolerability of ALX148, and to assess its efficacy in combination with venetoclax and azacytidine for AML treatment.

Regarding CD123, strongly expressed on LSCs, it has been reported that the murine CD123-specific 7G3mAb mediates ADCC by effector cells with specific effectiveness against leukemic cells. In NOD/SCID mice treated with 7G3, AMLLSC engraftment was deeply reduced, and survival improved. Two types of 7G3 mAbs are available. The first is chimeric (CSL360) and the second is humanized (CSL362), also known as talacotuzumab. The results of a phase I study of CSL360 in 40 patients with R/R or high-risk AML (NCT00401739) did not show a significant impact on the disease [82].

Given the inefficacy of CSL360, CSL362 was modified to bind, with higher affinity, both CD16A on NK cells by its Fc region and CD123 by its variable region, thereby more effectively mediating ADCC. As shown in Leukemia 2020, the use of talacotuzumab as monotherapy was clinically less effective in the high-risk myelodysplastic syndrome (HR-MDS) and AML groups of patients resistant to previous HMA therapy (NCT02992860). Significant toxicity compromised the success of the study and determined an early treatment discontinuation and disease progression [85]. Given the unfavorable risk/benefit profile of single-agent talacotuzumab, the addition of antibody–drug conjugates could determine an improvement in the anti-CD123 therapeutic approach. The results of a phase II/III study (NCT02472145) on the efficacy of talacotuzumab plus decitabine or decitabine alone in AML patients not eligible for chemotherapy showed no difference in efficacy between the two types of therapies. A total of 15% (12/80) and 11% (9/82) of patients receiving the combination and single-agent therapy achieved CR, respectively. The OS of patients undergoing combination therapy was 5.36 months, while that of patients treated with single-agent therapy was 7.26months [86]. In 2019, investigators produced the H9 mAb which shares the same CD123 antigen domain with the mAb CSL362 but recognizes a distinct epitope from that recognized by the CSL362 mAb. Importantly, they selectively killed AML primary leukemia cells and AML laboratory cell lines by ADCC [87].

### 2.4. Fc-Engineered Antibodies

Among engineered antibodies, there are those in which the Fc region is modified to enhance their antitumor activity by ADCC, antibody-dependent cellular phagocytosis (ADCP), and CDC. A variety of Fc-engineered mAbs are under investigation as potential effective therapies in hematologic malignancies. Besides CSL362, additional Fc-optimized antibodies are described below, based on evidence of their significant activity against cell surface AML-associated antigens.

The Fc-engineered CD33 antibody BI 836858 is one of them. Vasu et al. reported that it can mediate the killing of AML blasts by NK cells, which is further enhanced after decitabine pretreatment of AML blasts [88]. A clinical trial (NCT 02632721) is currently being undertaken to evaluate the effectiveness of BI 836858 in combination with decitabine in patients with AML. In addition, a phase I study (NCT03207191) investigated the safety and tolerability of the F16IL2 plus BI 836858 combination in AML relapse post-alloHSCT, but the results have not been published.

MEN1112, which is directed against CD157 and has recently entered clinical testing, is also an additional Fc-engineered antibody. Krupka et al. found that CD157 is often expressed in primary AML patient samples. Analysis by flow cytometry of 101 AML patient samples at primary diagnosis or relapse showed that CD157 is expressed in 97% of samples. It is more expressed at relapse than at the onset of the disease. Its persistent expression, from primary diagnosis to relapse, makes it an attractive candidate for targeted therapy at any stage of the disease [89]. An ongoing clinical trial (NCT02353143) is evaluating the effect of MEN1112, administered as an intravenous infusion, in patients with R/R AML.

## 3. Immune Checkpoint Inhibitors

The antitumor immune response is regulated by the type and the function of inflammatory cells infiltrating the tumor microenvironment. Numerous studies have clearly shown that CD8+ T cell infiltration is associated with a favorable prognosis in many solid tumors [90,91]. Nevertheless, upon immune pressure, malignant cells turn on mechanisms of evasion from T cell immunosurveillance. Thus, cancer cells may achieve this goal by a complex network of cooperation with immune cells and stromal cells. Then, anti-inflammatory (M2) tumor-associated macrophages (TAMs) are recruited into the tumor while regulatory T cells (Treg) are generated in the presence of anti-inflammatory molecules including the transforming growth factors (TGFs) and bone morphogenetic proteins (BMPs) [92,93,94,95,96]. Importantly, T cell activity can be further inhibited by stimulation of the immune checkpoint axes. The best-known immune [94] checkpoint axis is composed of inhibitory molecules expressed on T cells, such as the programmed cell death receptor-1 (PD-1(CD279)) and cytotoxic T lymphocyte-associated protein 4(CTLA4(CD152)), while the programmed cell death ligands 1(PD-L1(CD274)) and 2 (PD-L2(CD273)), B7-1 (CD80), and B7-2 (CD86) are expressed on the surface of cells mediating native immunity. Then, upon cell-to-cell conjugation, PD-1 and CTLA4 bind to PD-L1/PD-L2 and CD80/CD86, respectively. This interaction induces the deactivation of T cells, leading them to anergy. Unfortunately, malignant cells can also express the PD-1 and CTLA4 ligands, leading T cells to anergy. Immune checkpoint inhibitors prevent the interaction of “exhaustion” markers expressed by Tcells, with their ligands expressed by myeloid and/or tumor cells. By blocking negative regulators of Tcell immunity, such as cytotoxic CTLA-4, PD-1, and PD-L1, these biologicals can mediate antitumor immune responses. In AML, the best-proven immunotherapy is allogeneic HSCT [97], while the actual role of checkpoint inhibitors is less clear, although some positive results have been achieved in AML extramedullary relapse following HSCT [98]. Nevertheless, infiltration of T cells in the tumor milieu predicts responses to immune checkpoint inhibitors in AML [99,100].

Several checkpoint inhibitors have been approved for the treatment of solid tumors including the anti-CTLA-4 antibody ipilimumab and the anti-PD-1antibody pembrolizumab. Today, they have become the standard treatment in metastatic melanoma [101,102]. A PD-L1 inhibitor (atezolizumab) received FDA approval in 2016 for the treatment of patients with metastatic non-small cell lung cancer that has progressed during or after platinum-based chemotherapy [103].

The use of anti-PD-1 antibodies shows remarkable success both in Hodgkin and some non-Hodgkin lymphomas [104,105,106]. Furthermore, this typeof therapy might also be effective in leukemia, since antibodies against CTLA-4 and PD-1/PD-L1 have been shown to improve antileukemia immune responses in mice [107,108].The timing of treatment appears to play a fundamental role because leukemic cells show high proliferation rates and frequently manage to avoid the host’s immune responses. Therefore, the application of checkpoint inhibitors may achieve the best results in the presence of MRD and a complete immune system. Some clinical trials are now investigating the role of this type of antibody, alone or in combination with standard chemotherapy, in the treatment of primary AML or a post-transplant setting. CTLA-4 can interact with two natural ligands, CD80 (B7-1) and CD86 (B7-2), on antigen-presenting cells (APCs), competing with the costimulatory receptor CD28 and sending an inhibitory signal to T lymphocytes, preventing their maturation and differentiation. In AML, CD80 and CD86 are often overexpressed and may determine apoor outcome and a higher rate of relapse [109]. In these cases, anti-CTLA-4 antibodies such as ipilimumab could be very useful. Ipilimumab has shown good results in a phase I/Ib clinical trial conducted on 28 patients with relapsed hematologic cancer following HSCT. CR occurred in four patients with extramedullary AML and one patient with the MDS developing into AML (NCT01822509). It has been demonstrated that in relapsed hematological malignancies following allogeneic HSCT, leukemic cells downregulate the patient’s HLA haplotype and increase the expression of checkpoint inhibitor receptors. These results may unleash the cancer cell evasion from the donor T cell-mediated surveillance [98]. Moreover, preliminary data from a phase I study (NCT02890329), in which decitabine plus ipilimumab was utilized in patients with R/R MDS or AML following a bone marrow transplant or naïve treatment, were presented in the 62nd American Society of Hematology (ASH) Annual Meeting and Exposition, 2020. The findings indicated that ipilimumab, by targeting CTLA-4, enhances T cell-mediated malignant cell cytotoxicity. Promising response rates were also obtained by combining decitabine with ipilimumab.

PD-1 expressed on “exhausted” Tcells binds its ligand PD-L1 on APC. This link delivers intracellular signals inhibiting lymphocyte proliferation and activation. This pathway is normally used by the immune system to avoid response against self-antigens. In some tumors, upregulation of PD-1 andPD-L1 expression allows them to evade the host immune system. Antibodies against these molecules, such as nivolumab and pembrolizumab, are currently under investigation. As shown in the NCT02532231 phase II study, the use of nivolumab in high-risk AML patients in CR ineligible for SCT allowed reaching CR durations of 6 (79%) and 12 months (71%). Rates of OS were 86% at 12 months and 67% at 18 months [81]. Since PD1 inhibition alone demonstrated limited activity in AML [82], other researchers have evaluated the efficacy of PD-1 inhibitors in combination with ipilimumab or HMA such as azacytidine and decitabine in AML and MDS patients.

Nivolumab in combination with azacytidine was assessed in a phase IB/II trial in relapsed AML patients (NCT02397720); the first 53 patients enrolled were evaluated for response: 11 (21%) achieved CR/Cri, and 7 (14%) had hematologic improvement with an overall response rate of 35% [110].

At present, clinical results are promising, but responding patients are still few, and toxicity is considerable. Hence, there is a need to develop combination strategies to increase clinical benefits and reduce toxicities.

An effective therapeutic solution to treat patients that develop resistance to anti-PD-1 therapy could be represented by a combined blockade of PD-1 and other checkpoint inhibitors, including T cell immunoglobulin and mucin domain-containing molecule 3 (TIM-3).

TIM-3 is an immunoregulatory protein expressed on LSCs in most types of AML, but not on normal HSCs, which is able to enhance antitumor immunity and suppress tumor growth in several preclinical tumor models [111]. As reported by Goncalves Silva et al., this inhibitory receptor, together with its natural ligand galectin-9 (Gal-9), constitutes a secretory pathway that can be considered a potential target for AML immunotherapy [112].

Recent studies revealed that TIM-3 and PD-1 are coexpressed during exhausted T cell differentiation and influence T cell immunotherapy efficacy [113]. It seems that PD-1, binding to TIM-3/Gal-9, contributes to the reduction in TIM-3/Gal-9-induced cell death and the persistence of PD-1+TIM-3+ T cells [114]. A phase I clinical study based on a combinatory blockade strategy against PD-1 and TIM-3 (NCT03066648) is evaluating the safety and tolerability of an anti-TIM3 monoclonal antibody, named MBG453, as a single agent or in combination with the anti-PD1 antibody PDR001 (spartalizumab), and/or MBG453 in combination with decitabine, in patients with AML or high-risk MDS. The preliminary data indicate a good safety profile.

## 4. CAR-T Cell Therapy

Adoptive cell therapy (ACT) with tumor-infiltrating lymphocytes (TILs) or gene-modified T cells expressing novel T cell receptors (TCRs) or chimeric antigen receptors (CARs) is a rapidly emerging immunotherapy approach. Among the different types of ACT, the type that has made the most progress in clinical development is CAR-T cell therapy. It is based on the genetic engineering of a patient’s T lymphocytes to induce the expression of a chimeric receptor to be able to recognize a marker expressed on tumor cells, leading to cancer cell elimination. T lymphocytes are sampled from the patient’s peripheral blood and transduced with viral vectors encoding the desired genes. Genetically engineered lymphocytes are then led to proliferating in vitro before re-infusion into the patient’s blood (Figure 2).

Severe immune-mediated adverse events following CAR-T cell infusion have been reported; therefore, unwanted toxicity management is one of the key points in implementing this type of ACT. Good selection of the optimal target antigens is essential to overcome the limitations of this approach. Thus far, CD19 has been the main target antigen for CAR-T cell therapy, and encouraging results have been reported, especially in the treatment of B cell neoplasms [88]. In 2013, the use of CD19-directed CAR-T (CART19) cells in two children with ALL at the Children’s Hospital of Philadelphia marked the first success of this branch of immunotherapy. Despite severe adverse events noted in both children, complete remission was reached [115]. Using this therapy to treat AML is more complicated because of the non-restricted expression of AML-associated antigens.

Nevertheless, several single-chain variable fragment (scFv) antibodies specific for AML antigens have been utilized to produce CAR-T cells for clinical investigation. Among them, the single-chain anti-CD33 (CART-33) seems to be the most relevant. However, severe side effects including neurotoxicity and CRS have been experienced. In addition, in a clinical trial (NCT01864902), CART-33 did not affect the clinical course of refractory AML, leading to progression within nine weeks following CART-33 cell infusion. Thus, the trial has been suspended by the US FDA for two months until the adoption of more severe criteria. For the treatment of R/R AML patients, the authors proposed the use of CART-33 infusions as a short-term problem-solving approach followed by chemotherapy or HSCT [116].

However, only a few clinical studies of CAR-T cell immunotherapy for AML patients are currently being undertaken (Table 2). This may be due to limitations associated with the poor clinical outcome detected thus far with anti-AML CARs, which includes the CAR-T cell unresponsiveness due to the strong immunosuppressive environment in the bone marrow of AML patients, and the possible generation of myeloid malignant cell escape variants following antigen loss. Additional outstanding clinical issues involve the development of strategies capable of overcoming the cancer cell immune evasion mechanisms from CAR-T cell immunosurveillance [117], and how to increase CAR-T cell-related responses with the administration of antibodies directed towards immune inhibitory checkpoint molecules [118,119].

## 5. Vaccine-Based Therapies

Following HSCT, the powerful graft-versus-leukemia (GVL) effect, mediated by allogeneic donor lymphocytes, has proven to be the most impressive evidence of how the immune system can control AML spread. However, given the low range of specificity of the donor lymphocytes, the recipient may develop graft-versus-host disease (GVHD), which is linked to increased morbidity and mortality [120]. An attractive strategy aimed at generating a specific immune response is that of producing specific T lymphocytes. Despite the fact an ideal AML-associated antigen has not been identified yet, several potential antigens have been described. Among them, there are the leukemia-specific antigens (LSAs), including myeloid primary granule proteins [121,122], leukemia-associated antigens (LAAs), cancer/testis antigens (CTAs), and ubiquitous antigens. However, at present, only the first two are used for vaccine construction. In this context, vaccines can be designed by using peptides presented by professional antigen-presenting cells (APCs), such as dendritic cells (DCs). The first vaccine against AML was created in the late 1960s.Calmette–Guérin strain mycobacteria (BCG) and irradiated AML cells were combined to stimulate the immune system as maintenance therapy, but this method did not provide the expected results. Only one out of four trials carried out in 41 patients showed increased survival (median: 90 versus 45 weeks) and remission duration (median: 35 versus 20 weeks), and overall clinical results were not confirmed by others [123,124,125,126]. Despite this daunting start, the search for a strategy based on vaccines for the treatment of AML continues to be actively pursued. The goal would be to prevent relapse by attacking those cells, such as stem cells, refractory to chemotherapy. LSAs, such as the promyelocytic leukemia-retinoic acid receptor α (PML-RARα), are an intriguing target; unfortunately, they are expressed in a minority of AML patients. LAAs are also expressed on normal cells but can be overexpressed in leukemia cells. Therefore, they are candidates for immunotherapy [127,128]. Wilms’ tumor 1 (WT1) antigen, proteinase (PR)-1 and -3, and receptor for hyaluronic acid-mediated motility (RHAMM) are the most used as peptide vaccines [109,122,129,130,131]. These peptides are inoculated preferentially in micellar delivery systems to obtain a slower release to APC. APCs then more effectively present peptides to T cells. Several studies demonstrated the advantages of this therapy. In 2015, Di Stasi et al. showed that WT1 vaccination is safe, feasible, and potentially effective in patients with AML [132]. Consistent with prior WT1 vaccine studies, a phase I pilot study proved that WT1-directed peptide vaccination was effective and well-tolerated in 16 heavily pretreated AML and MDS patients [133].

These data were also confirmed by a phase I study (NCT01266083) that investigated the use of a multivalent WT1 peptide vaccine (galinpepimut-S) in adults with AML in the first complete remission [134]. However, immune responses are limited to specific patients, depending on their HLA typing, and are often of short duration [12]. To overcome these limitations, innovative mixtures of short and long heteroclitic peptides with higher HLA affinity and that are able to trigger not only CD8+but also CD4+ Tcells have been developed [11]. Moreover, improvements could also be obtained using new adjuvants that function as immune potentiators.

Vaccination could also be performed using DCs, potent APCs able to induce strong specific antitumor immune responses. DCs can be obtained from autologous or allogeneic leukapheresis [135] and by “ex vivo” differentiation. They are then loaded with tumor antigens and re-infused in the patient to trigger antitumor immune responses (Figure 3).

Clinical trials have shown that DC-based immunotherapy is feasible, safe, and devoid of serious side effects [136,137,138].

As reported in Van Acker 2019, major efforts are concentrated on the use of DCs derived from autologous peripheral blood monocytes (moDCs) [138] or autologous leukemic blast cells (AML-DCs) [139]. Emerging results show that moDCs enhance activation of autologous leukemia-specific T cells more effectively than AML-DCs. This could be explained by the lack of 4-1BB ligand (4-1BBL) expression on AML-DCs [140]. Alternatively, the expression of indoleamine 2,3-dioxygenase 1 (IDO-1) by leukemic blasts could promote a more tolerogenic microenvironment [141]. However, it is important to keep in mind that the AML-DCs can present the entire antigenic repertoire of AML blasts. Instead, moDCs must be loaded with AML antigens. This can be achieved by exogenous pulsing with specific peptides such as WT1, apoptotic AML blasts, and blast lysates. Additional methodologies comprise AML blast–DC fusion and messenger RNA (mRNA) electroporation. The latter technique was used in a phase II study carried out in 2013. It investigated the impact of DC vaccination on the prevention of relapse in 29 patients with AML, 26 in CR, and 3 in PR [142]. Immunization was provided by the intradermal administration of a vaccine obtained with DCs derived from blood monocytes electroporated with mRNA encoding the WT1. An antileukemic effect was demonstrated in 8 out of 29 patients. Clinical studies using monocyte-derived DCs loaded with various antigens are ongoing in several countries. Preliminary results are encouraging, and the use of new combinations may improve the efficacy of DC vaccination and further enhance antitumor immune responses.

## 6. Future Perspectives

The identification of novel therapeutic methods capable of modulating the expression of relevant genes in human diseases is challenging. A major hurdle for successful immunotherapy of AML is the lack of optimal AML antigens to be targeted. In this context, a recently developed area of investigation is epitranscriptomics. This area focuses on the changes occurring in the cells following post-transcriptional RNA modification. More than 170 RNA modifications are known. These modifications impact RNA functionality in collaboration with RNA binding proteins termed writers, readers, and erasers. Writer enzymes “write” the RNA modification. The readers read and interpret them, and the erasers remove them. The changes can affect both messenger RNA (mRNA) and ribosomal RNA (rRNA) as well as transfer RNA (tRNA). One of the most common modifications is N6-methyladenosine (m6A) methylation; it can carry out relevant biological functions that affect hematopoietic malignancies. The writer methyltransferase-like 3 (METTL-3) and its associated methyltransferase-like 14 (METTL-14), and the reader YTH domain-containing family protein 2, are critical for LSC survival. Similar considerations can be applied to the demethylases fat mass and obesity-associated (FTO) protein, and AlkB homolog 5 RNA demethylase. Additionally, N6-methyladenosine (m6A) methylation can produce significant effects in the regulation of innate and adaptive immune responses including tumor immunology and immunotherapy [143].

This scenario opens new perspectives in the treatment of AML. However, the state of the art of current knowledge represents the platform for further investigation toward a better understanding of the role of both them6A and other RNA modifications in the identification of optimal AML antigens for innate and/or adaptive immunotherapy of AML.

## 7. Conclusions

AML continues to be a severe disease. The identification of novel immunotherapeutics, among them GO and tagraxofusps, represents an additional step ahead in the fight against AML. Therefore, there is a hope to increase the availability of these molecules in a way to reduce the use of chemotherapy to enhance the selectivity of the treatment and reduce the patient side effects. Additionally, these new treatments are easy to handle, providing investigators with the possibility to administer the treatment in an outpatient setting.

Although this is only the beginning, these new drugs are showing never-before-seen results, saving patients who would not have had survival chances without them.

In conclusion, the latest immunotherapeutic drugs allow more patients to experience a better clinical course of the disease.

## Figures and Tables

**Figure 1 cancers-13-06246-f001:**
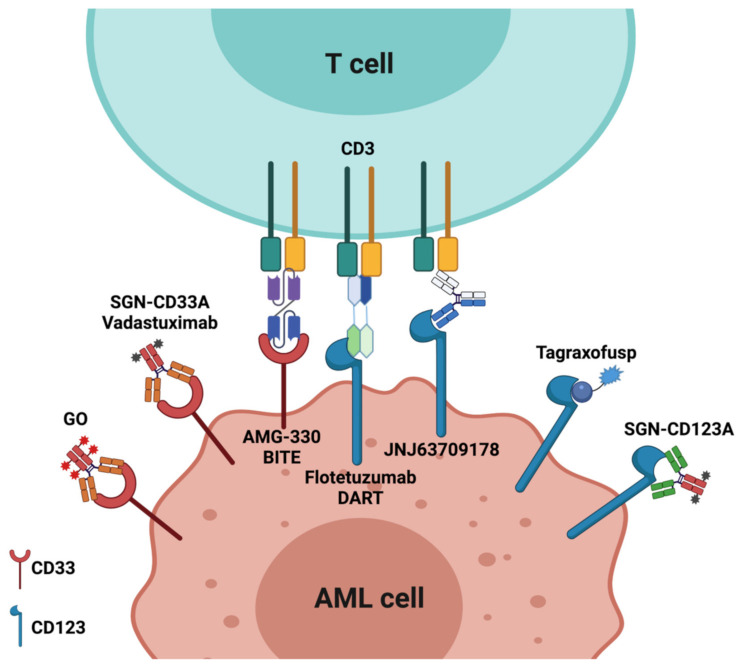
Immunotargeting of validated targets in AML: approved versus investigational molecules. CD33 targeting: GO (Gentuzumabozogamicin) for favorable and intermediate AML in association with daunorubicine and cytarabine; SGN-CD33A and AMG-330 under investigation. CD123 targeting: tagraxofusp approved; SGN-123A, flotetuzumab, and JNJ63709178 investigational. The figure was created with BioRender.com.

**Figure 2 cancers-13-06246-f002:**
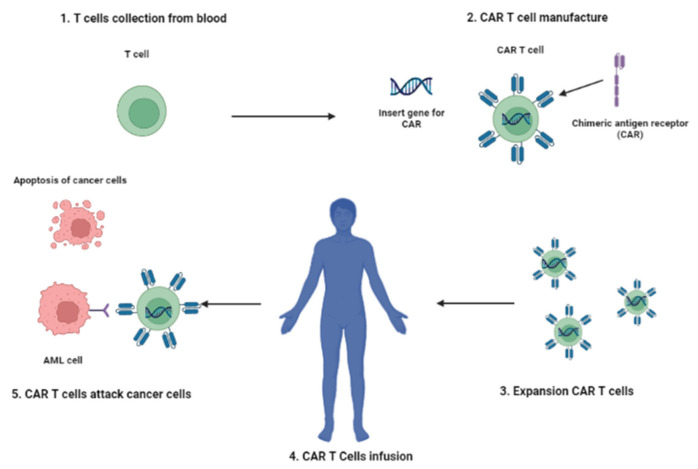
CAR-T immunotherapy. (**1**) Patient’s T cells are collected by leukapheresis. (**2**) A viral vector delivers a gene encoding a CAR into the T cells. (**3**) Expansion of CAR-expressing T cells. (**4**)The CAR-T cells are infused into the patient’s blood (**5**). CAR-T cells attack cancer cells. The figure was created with BioRender.com.

**Figure 3 cancers-13-06246-f003:**
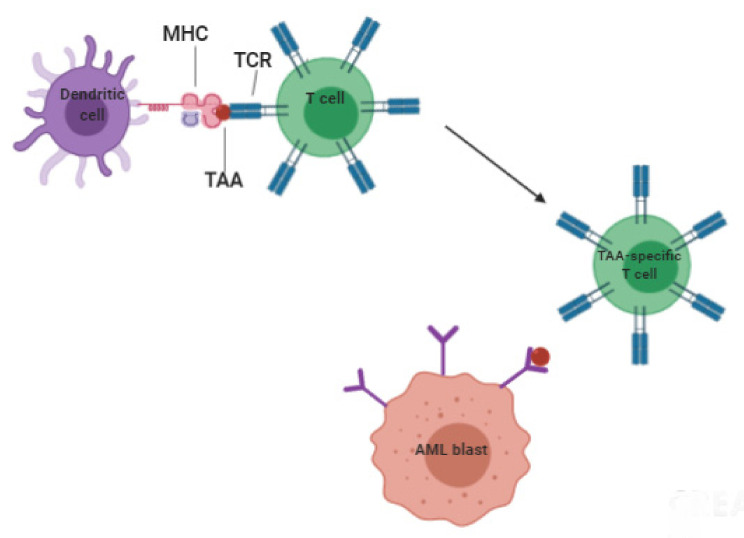
Schematic representation of T cell responses toward AML tumor-associated antigen (TAA). DCs present the MHC, loaded with AML antigen peptides, to the TCR of T lymphocytes, allowing the generation of AML-specific Tcell activation and expansion. Among AML-specific T cells, CTL will be able to recognize and kill AML cells. The figure was created with BioRender.com.

**Table 1 cancers-13-06246-t001:** Clinical trials of bispecific antibodies for leukemia patients.

Drug	Target	Phase	ClinicalTrials.gov Identifier	Estimated Enrollment Number	Disease Conditions	Status
AMG 330	CD33/CD3	1	NCT02520427	256	R/R AML/MRD Positive AML/MDS	Active/Recruiting
MGD006	CD123/CD3	1/2	NCT02152956	330	Primary Induction Failure (PIF) or Early-Relapse (ER) AML	Active/Recruiting
JNJ-63709178	CD123/CD3	1	NCT02715011	62	R/R AML	Recruitment Completed
MCLA 117	CLL1/CD3	1	NCT03038230	62	R/R AML	Active/NotRecruiting

**Table 2 cancers-13-06246-t002:** Ongoing clinical trials of CAR-Tcell immunotherapy.

Target	Phase	ClinicalTrials.gov Identifier	Estimated Enrollment Number	Status	Disease Conditions	Intervention/Treatment
CD33, CD38, CD123, CD56, MucI, CLL1	1/2	NCT03222674	10	Unknown	R/R AML	Muc1/CLL1/CD33/CD38/CD56/CD123-specific gene-engineered T cells
CD33, CD38, CD56, CD117, CD123, CD34, Muc1	1	NCT03291444	30	Active/Recruiting	ALL/R/R AML/MDS	CAR-T cells/Eps8or WT1 peptide-specific dendritic cells
CD123	1	NCT02159495	42	Active/Recruiting	R/R AML or BPDCN	Cyclophosphamide/autologous or allogenic CD123CAR-CD28-CD3zeta-EGFRt-expressing T lymphocytes/fludarabine phosphate
CD123	1	NCT03114670	20	Unknown	Adult relapsed AML following allogeneic HSCT	CD123CAR-41BB-CD3zeta-EGFRt-expressing T cells
CD123	1	NCT03190278	65	Active/Recruiting	R/R AML	UCART123v1.2(allogeneic engineered Tcells expressing anti-CD123 chimeric antigen receptor)
CD123	1/2	NCT03556982	10	Unknown	R/R AML	Fludarabine-cyclophosphamide chemotherapy followed by infusion of allogeneic or autologous CD123-targeted CAR-T cells
CD123	1	NCT03766126	12	Active/Not recruiting	Adult R/R AML	Fludarabine-cyclophosphamide chemotherapy followed by infusion of anti-CD123 CAR-T (autologous lentivirally transduced) (CD123CAR-41BB-CD3)
CD123,CLL1	2/3	NCT03631576	20	Active/Recruiting	R/R AML	CD123/CLL1 CAR-T cell therapy
CD123	1	NCT03796390	15	Unknown	R/R AML	Chemotherapy/CD123 CAR-T cells (autologous lentivirally transduced)
CD44	1/2	NCT04097301	58	Active/Recruiting	R/R AML, MM	CD44v6 CAR-Tcells (MLM-CAR44.1 Tcells), cyclophosphamide, and fludarabine

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
