# Peer review of "Immunotherapy as a Turning Point in the Treatment of Acute Myeloid Leukemia"

_cancers, 2021, doi:10.3390/cancers13246246_

Round 1
Reviewer 1 Report
This manuscript is a comprehensive review about the progress of immunotherapy against AML in which antibody-based therapy, immune checkpoint inhibitors, adoptive cell therapy and vaccine-based therapies. The manuscript is well-written with substantial information about past and ongoing clinical trials for each treatment strategies which are mostly well referenced. Minor comments:
(1) Line 70 says “the majorable residual disease (MRD)”, do you mean minimal/measurable residual disease?
(2) Line 474 says "Nivolumab in combination with azacytidine was assessed...". Reference for this fact is not seen in that paragraph.
(3) For 2.4, it would be better to simply explain what Fc-engineered antibodies are and what advantages they have over other engineered antibodies.
(4) Line 608 says "The latter technique has been used in a phase II study..." which lacks reference.
Author Response
Point-by-point response to reviewer's comments
Comment 1-Line 70 “the majorable residual disease (MRD)”, do you mean minimal/measurable residual disease?
Author reply : reads “minimal/measurable residual disease”. Line 70.
Comment 2-Line 474 "Nivolumab in combination with azacytidine was assessed...". Reference for this fact is not seen in that paragraph.
Author reply: The reference has been added as requested [112]. Line 476.
Comment 3-For 2.4, it would be better to simply explain what Fc-engineered antibodies are and what advantages they have over other engineered antibodies.
Author reply: At the beginning of 2.4 section, we described Fc-engineered antibodies and their advantages over other engineered antibodies. Lines 378 to 383.
Comment 4-Line 608 says "The latter technique has been used in a phase II study..." which lacks reference.
Author reply: the reference has been added [143]. Line 614.
Reviewer 2 Report
The article written by Aureli et.al, entitled “Immunotherapy as a turning point in the treatment of acute myeloid leukemia” is well-organized, which covers most of the latest information related to AML immunotherapy. The way of sequential categorization of AML therapy is nicely organized, for example, antibody-based therapy (monoclonal antibody and bi-specific antibodies), Immune checkpoint inhibitors, vaccines, and CAR T-cell therapy. However, few important points are missing in this review; therefore, it would be impressive to consider my suggestions to further improve the quality of the manuscript.
Comment-1: The author should refer to this most important publication relevant to AML immunotherapy. “Immunotherapeutic Potential of m6A-Modifiers and MicroRNAs in Controlling Acute Myeloid Leukaemia”, 2021, PMID: 34207299. In addition to the already mentioned different types of therapies for AML, the author should consider adding new section about “Epitranscriptomics” or RNA modifying drugs (RMDs) and microRNAs in AML therapy. An additional table mentioning these RMDs in AML therapy would improve the visibility of the publication.
Comment-2: In section 2.3. “Monoclonal antibodies directed against human leukemia stem cells (LSCs)” is a sub-heading of main section 2. “Antibody-based therapy”. So it should not be bold. Make it similar as other subheading ‘font’.
Comment-3: Abstract section has different font size, so it should be normalized.
Comment-4: Reference 2 and 3 should be join together.
Comment-5: In addition to section-5. Vaccine-based Therapies, DC vaccination can be added.
Comment-6: Some of the sub-sections are unnecessarily elaborated. So make it concise/to-the-point to create interests to the readers.
Author Response
Point-by-point response to the reviewer's comments
Comment-1: The author should refer to this most important publication relevant to AML immunotherapy. “Immunotherapeutic Potential of m6A-Modifiers and MicroRNAs in Controlling Acute Myeloid Leukaemia”, 2021, PMID: 34207299. In addition to the already mentioned different types of therapies for AML, the author should consider adding new section about “Epitranscriptomics” or RNA modifying drugs (RMDs) and microRNAs in AML therapy. An additional table mentioning these RMDs in AML therapy would improve the visibility of the publication.
Author reply: the reference has been added [144]. Line 644.
A new section (6. Future perspectives) including epitranscriptomics to evaluate the new therapeutic approaches in AML treatment has been added in the review. We have chosen to deepen this topic in our next papers in order to provide the reader with more detailed information Lines 622 to 649.
Comment-2: In section 2.3. “Monoclonal antibodies directed against human leukemia stem cells (LSCs)” is a sub-heading of main section 2. “Antibody-based therapy”. So it should not be bold. Make it similar as other subheading ‘font’.
Author reply: Font of section 2.3 has been changed. Line 287.
Comment-3: Abstract section has different font size, so it should be normalized.
Author reply: Font size of the abstract has been normalized.
Comment-4:Reference 2 and 3 should be join together.
Author reply: the references 2 and 3 are joined together. Line 44.
Comment-5: In addition to section-5. Vaccine-based therapies, DC vaccination can be added.
Author reply: In section 5, DC vaccination is better described in text. Lines 588 to 620.
Comment-6: Some of the sub-sections are unnecessarily elaborated. So make it concise/to-the-point to create interests to the readers.
Author reply: To this ends, we have made some efforts to fulfill this request.
Reviewer 3 Report
In this review article, Aureli et al. comprehensively documented a recent progress in treatment of AML, mainly based on the results of clinical studies. Therefore, this review should provide readers with up-to-date information regarding newly developed ways for AML treatments. I found some typos in the text; for example, page 1, line 43. Therefore, the authors should carefully check the text in the final version of the manuscript.
Author Response
Point-by-point response to the reviewer's comments
In this review article, Aureli et al. comprehensively documented a recent progress in treatment of AML, mainly based on the results of clinical studies. Therefore, this review should provide readers with up-to-date information regarding newly developed ways for AML treatments. I found some typos in the text; for example, page 1, line 43. Therefore, the authors should carefully check the text in the final version of the manuscript.
Author reply: we enclosed a new paragraph (6. Future perspectives) including epitranscriptomics to evaluate the new therapeutic approaches in AML treatment. Lines 622 to 649.
Text in the final version of the manuscript has been carefully checked.
Reviewer 4 Report
The manuscript is well written and gives a comprehensive overview on the development of immunotherapeutic strategies to treat AML.
I have just one comment on the CD47/SIRP part. There are two Blood publications by Hans-Joerg Bühring who was the first to demonstrate that Sirp alpha but not beta binds to CD47. In addition he showed that SIRP-a is expressed on hematopoetic cells and dendritic cells. This references should be included. In addition, there was just recently published a report demonstrating that CD47/Sirp a is involved in NK cell function and is another additional mechanism for tumor rejection (Deuse et al., JEM, 2021).
There are several other CD47 and Sirp-a antibodies that are currently used in clinical trials that should be mentioned.
Author Response
Point-by-point response to the reviewer's comments
1-I have just one comment on the CD47/SIRP part. There are two Blood publications by Hans-Joerg Bühring who was the first to demonstrate that Sirp alpha but not beta binds to CD47. In addition he showed that SIRP-a is expressed on hematopoetic cells and dendritic cells. This references should be included. In addition, there was just recently published a report demonstrating that CD47/Sirp a is involved in NK cell function and is another additional mechanism for tumor rejection (Deuse et al., JEM, 2021).
1-Author reply: Publications have been included as requested [74-76]. Lines 322-323.
2-There are several other CD47 and Sirp-a antibodies that are currently used in clinical trials that should be mentioned.
2-Author reply: further CD47 and Sirp-a antibodies have been enclosed in this section. Lines 335 to 349.
Round 2
Reviewer 2 Report
The author has carefully taken my suggestions and implemented them in the revised manuscript. The manuscript is now looking much improved. However, in ‘Section 2.2 Bispecific Antibodies’ the author has described well about bispecific antibodies and mentioned CD33/CD3BiTE, named AMG330 with reference 48 in line# 242. Moreover, the author has also described different types of antibodies like DART in line #254. In addition, some phase-I clinical trial was also mentioned in line# 270 and table-1. Next, section 4. CAR T- cell therapy, line# 525-530 describes the most recent AML treatment by CAR T-cell therapy. I feel these subsections need appropriate supporting references. Therefore, the author is suggested to cite this important article published recently in 2021. Several clinical trials and details of advanced AML therapy are discussed extensively in this paper, and thus it deserves to be cited.
https://stm.bookpi.org/RDMMR-V14/article/view/4991
Kumar, S., Ashraf, M. U., Aman, A. K., & Bae, Y.-S. (2021). Advances in Personalized Therapy: Co-targeting Intracellular Immune checkpoints in Controlling Acute Myeloid Leukemia. Recent Developments in Medicine and Medical Research Vol. 14, 108–151. https://doi.org/10.9734/bpi/rdmmr/v14/14436D
Author Response
Reference has been added. [122] Line 545.